# An 18S V4 rRNA metabarcoding dataset of protist diversity in the Atlantic inflow to the Arctic Ocean, through the year and down to 1000 m depth

Elianne Egge[1,2], Stephanie Elferink[3], Daniel Vaulot[4,5], Uwe John[3], Gunnar Bratbak[6], Aud Larsen[7], and Bente Edvardsen[1]

[1]University of Oslo, Department of Biosciences, Section for Aquatic Biology and Toxicology, PO Box 1066 Blindern, NO-0316 Oslo, NORWAY

[2]University of Duisburg-Essen, Fakultät für Biologie, Universitätsstr. 5, DE-45141 Essen, GERMANY (present address)

[3]Alfred-Wegener-Institut Helmholtz-Zentrum für Polar- und Meeresforschung. Am Handelshafen 12, Bremerhaven DE-27570, GERMANY

[4]UMR7144, CNRS, Sorbonne Université, Station Biologique de Roscoff. Place Georges Teissier, FR-29682 Roscoff, FRANCE

[5]Asian School of the Environment, Nanyang Technological University, 50 Nanyang Avenue, Singapore 639798 SINGAPORE

[6]University of Bergen, Department of Biological Sciences, PO Box 7803, NO-5020 Bergen, NORWAY

[7]NORCE Norwegian Research Centre, PO Box 7810, NO-5020 Bergen, NORWAY

**Correspondence:** Elianne Egge (elianne.egge@gmail.com)

**Abstract.** Arctic marine protist communities have been understudied due to challenging sampling conditions, in particular during winter and in deep waters. The aim of this study was to improve our knowledge on Arctic protist diversity through the year, both in the epipelagic (< 200 m depth) and mesopelagic zones (200-1000 m depth). Sampling campaigns were performed in 2014, during five different months, to capture the various phases of the Arctic primary production: January (winter), March (pre-bloom), May (spring bloom), August (post-bloom) and November (early winter). The cruises were undertaken west and north of the Svalbard archipelago, where warmer Atlantic waters from the West Spitsbergen Current meets cold Arctic waters from the Arctic Ocean. From each cruise, station, and depth, 50 L of sea water were collected and the plankton was size-fractionated by serial filtration into four size fractions between 0.45-200 $\mu$m, representing picoplankton (0.45-3 $\mu$m), small and large nanoplankton (3-10 and 10-50 $\mu$m, respectively) and microplankton (50-200 $\mu$m). In addition, vertical net hauls were taken from 50 m depth to the surface at selected stations. The net hauls were fractionated into the large nanoplankton (10-50 $\mu$m) and microplankton (50-200 $\mu$m) fractions. From the plankton samples DNA was extracted, the V4 region of the 18S rRNA-gene was amplified by PCR with universal eukaryote primers and the amplicons were sequenced by Illumina high-throughput sequencing. Sequences were clustered into Amplicon Sequence Variants (ASVs), representing protist genotypes, with the dada2 pipeline. Taxonomic classification was made against the curated Protist Ribosomal Reference database (PR$^2$). Altogether, 6,536 protist ASVs were obtained (including 54 fungal ASVs). Both ASV richness and taxonomic composition varied between size-fractions, seasons, and depths. ASV richness was generally higher in the smaller fractions, and higher in winter and the mesopelagic samples than in samples from the well-lit epipelagic zone during summer. During spring and summer, the phytoplankton groups diatoms, chlorophytes and haptophytes dominated in terms of relative read abundance in the epipelagic

zone. Parasitic and heterotrophic groups such as Syndiniales and certain dinoflagellates dominated in the mesopelagic zone all year, as well as in the epipelagic zone during the winter. The dataset is available at https://doi.org/10.17882/79823 (Egge et al., 2014).

## 1   Introduction

The West Spitsbergen current is considered the main gateway from the Atlantic into the Arctic Ocean, as it flows along the west side of the Svalbard Archipelago, transporting relatively warm and salty water (T > 2°C, S > 34.92; c.f. Randelhoff et al. (2018)) into the Barents Sea and Arctic Ocean (Figure 1). In response to global warming, this current has become both warmer and stronger in recent years, increasingly replacing water advected from the central Arctic Ocean with warm and salty water of Atlantic origin, a process referred to as "Atlantification" (Årthun et al., 2012). This increase in oceanic heat in the Arctic area correlates with the rapid decline in ice extent observed over the past decades (Årthun et al., 2012). Increased inflow of Atlantic water affects the primary production and protist communities in several ways. Water mixing happens more easily in the Atlantic water, because in contrast to the permanently salinity-stratified central Arctic Ocean, the water column is temperature-stratified and less stable, thus upper-ocean nutrients are more efficiently replenished early in winter. Furthermore, the warm Atlantic water melts the ice, and a layer of fresh, cold water is formed near the surface. The timing of this stratification is crucial for the onset of the spring bloom, and thinner ice means less light-limitations for the algae living inside and under the ice. However, the loss of sea ice also results in the loss of habitat for many protists, especially those adapted to a life in or on the ice. These various effects of climate change may thus alter both the location and timing of blooms, as well as their biomass and species composition (e.g. Eamer et al., 2013; Li et al., 2009).

To understand what consequences environmental changes in this Arctic region will have for the biodiversity of the whole pelagic community and for the production through the food web up to higher trophic levels, we need to know what are the community components and where and when the organisms occur. This will also enable us to detect future changes. However, still relatively little is known about the diversity and distribution of protists in the Arctic Ocean (e.g. Lovejoy, 2014). Arctic winter microbial eukaryote communities are particularly understudied due to logistic challenges which include ice cover and frequent storms. Metabarcoding using high-throughput sequencing has become a commonly used method to study the community composition of marine protists, and has revealed a huge unknown diversity (e.g. de Vargas et al., 2015). In recent years, several metabarcoding studies of protist communities in the Arctic Ocean have been undertaken, but most represent only snapshots of the community as based on a single cruise or season (e.g. Bachy et al., 2011; Kilias et al., 2014; Monier et al., 2015; Vader et al., 2015). Studies that have sampled the full yearly cycle have typically only sampled the upper water column (0-50 m depth) (e.g. Marquardt et al., 2016).

Here we present a metabarcoding dataset from the Northern Svalbard region of the Arctic Ocean sampled during five cruises representing the full seasonal cycle, and at 3-4 depths from the surface down to 1000 m. Metabarcoding targeted the V4 region of the 18S rRNA gene. The data are provided both as raw reads and as Amplicon Sequence Variants obtained after processing with the dada2 pipeline, with corresponding ASV abundance tables. The data presented here were obtained within the

framework of the project 'MicroPolar' (https://www.researchinsvalbard.no/project/7280). The virus and prokaryote communities from the same project have been described in Sandaa et al. (2018), and Wilson et al. (2017) and Paulsen et al. (2016), respectively. Environmental data from the MicroPolar sampling campaign have previously been published in Paulsen et al. (2017) and Randelhoff et al. (2018). A subset of the environmental data corresponding to the stations and depths of the protist metabarcoding samples is included in the data repository of the present study (https://doi.org/10.17882/79823/).

## 2 Study area and general environmental conditions

The physical and biogeochemical oceanographic conditions during these cruises have previously been described in detail in Randelhoff et al. (2018). Flow-cytometric counts of viruses, bacteria and pico- and nanoplankton have been published in Sandaa et al. (2018). We briefly describe the methods and reiterate the main results here to provide background for our metabarcoding data. The complete data set from Randelhoff et al. (2018) can be found at the PANGAEA Data Publisher for Earth and Environmental Science (https://doi.pangaea.de/10.1594/PANGAEA.884255), (Paulsen et al., 2017). The reader is advised to consult the original papers for detailed descriptions of the methods. The environmental parameters included here are listed in Table 1.

### 2.1 Study area

Sampling campaigns were performed in 2014 as described in Paulsen et al. (2017); Wilson et al. (2017); Randelhoff et al. (2018); Sandaa et al. (2018), during five different months, to capture the various stages of the Arctic primary production: January (06.01–15.01, winter), March (05.03-10.03, pre-bloom), May (15.05-02.06, spring bloom), August (07.08-18.08, post-bloom) and November (03.11-10.11, early winter). The cruises were undertaken west and north of the Svalbard archipelago, where warmer Atlantic water in the West Spitsbergen Current meets colder water from the Arctic Ocean (Figure 1). Bottom depth varied from 327 m (November station N03) to c. 3000 m (March station M05). The area and locations for each sampling campaign were as similar as possible, but constrained by the sea ice cover, from 79 to 82.6 °N. During each cruise, transects of 3-6 stations were sampled at three or four depths: in the epipelagic zone at 1 m and at the deep chlorophyll maximum (usually between 15-25 m), and in the mesopelagic zone at one or two depths, as a rule 500 m and 1000 m, or as deep as the bathymetry of the station permitted.

### 2.2 Environmental conditions

#### 2.2.1 Daylength and euphotic zone depth

Daylength at each cruise and station was calculated with the 'daylength' function in the 'geosphere' R package Hijmans (2019). Daylength was 0 hours during winter (January, November), 6-8 hours in March, and 24 hours in May and August. Continuous profiles of photosynthetically available radiation (PAR; radiation at wavelengths between 400 and 700 nm) in the upper ocean were measured during the May and August cruises using a RAMSES radiometer (TriOS, Germany) with a wavelength spectrum

of 190–575 nm. The euphotic zone depth ($Z_{eu}$) was then defined as the depth at which downwelling PAR reached 1 % of its value just below the surface. Euphotic zone depth was 19-23 m in May, 22 m in August at station P05, and 48 and 45 m at station P06 and P07, respectively. The uncertainty of these values is 2-3 m (Randelhoff et al., 2018).

### 2.2.2  Ice cover

The ice extent was smallest in January, and peaked in May (see Figure 1 in Wilson et al., 2017). The two stations sampled in January were in the open ocean, whereas in March, May and August, all the stations were situated in varying degrees of drift ice, except March station M06 and August station P05, which were situated in open water. In November, all stations were in open water, except November station N02, which was in open drift ice (see Wilson et al., 2017, for the definition of the different ice types).

### 2.2.3  Hydrographical conditions

Vertical profiles of temperature, salinity, and fluorescence were recorded at each sampling station using an SBE 911plus CTD system (Sea-Bird Scientific USA, Bellevue, WA, USA). Conditions were dominated by the large-scale inflow of warm Atlantic Water (the West Spitsbergen current), which is modified as it enters the cold Arctic Ocean. Surface temperature was highest in August, station P05, $\simeq$ 6 °C. Surface temperature and salinity were generally lower at the stations farther off the slope compared to those on the shelf slope (Figure 2). The difference between stations diminished by depth, and at 1000 m the conditions were almost identical across stations and months (Figure 2).

### 2.2.4  Inorganic nutrients and Chlorophyll *a*

Water samples for nutrients and Chl *a* were taken with 8 L Niskin bottles mounted on a General Oceanics 12-bottle rosette. Nutrients ($NO_2^-$ +$NO_3^-$ , $Si(OH)_4$, $PO_4^{3-}$) were frozen until analysis and analyzed by standard seawater methods using a Flow Solution IV analyzer from O.I. Analytical, USA. Atlantic water was the dominant source of nutrients (as indicated by $PO_4^{3-}$:$NO_3^-$, c.f. Randelhoff et al. (2018)). In the surface, inorganic nutrients and Chl *a* were inversely related to each other (Figure 2). As expected, Chl *a*-concentrations were close to zero in the dark winter months (November and January). In March, there was some daylight, but the water column was not yet stratified, which prevented initiation of the spring bloom. Chl *a* concentrations peaked in May (at most 14 $\mu$gL$^{-1}$), concomitantly with depletion of inorganic nutrients. From May to August Chl *a* concentration decreased to < 5 $\mu$gL$^{-1}$, while the concentrations of inorganic nutrients were still generally low. By November, the concentrations of inorganic nutrients in the epipelagic zone had increased and were again back to the levels observed in January and March.

### 2.2.5  Cell counts

FCM analysis were performed using an Attune Focusing Flow Cytometer (Applied Biosystems by Life Technologies) with a syringe-based fluidic system and a 20mW488 nm (blue) laser. The samples were fixed with glutaraldehyde (0.5 % final

conc.) at 4 °C for a minimum of 30 min, flash frozen in liquid nitrogen and stored at -80 °C until analysis. Autotrophic phytoplankton was categorised as the cyanobacterium *Synechococcus*, picophytoplankton or nanophytoplankton, based on variation in side scatter, chlorophyll *a* and phycoerythrin autofluorescence. Samples for enumeration of virus-like particles, heterotrophic prokaryotes and nanoflagellates were stained with SYBR Green I, and distinguished based on side scatter and green fluorescence. Virus-like particles were catgorised as "small", "medium" and "large", based on fluorescence intensity. Cell counts of pico- and nanophytoplankton, heterotrophic nanoflagellates, *Synechococcus* and heterotrophic prokaryotes were generally higher in May and August than during winter and early spring, and higher in the epipelagic than the mesopelagic samples. Virus counts were generally higher in January and August than the other months, and the variation by depth was smaller than for prokaryotes and eukaryotes (c.f. Figure 3 and 4 in Sandaa et al. (2018)).

## 3   Sampling strategy

### 3.1   Sample preparation for DNA extraction

#### 3.1.1   Niskin bottles

From each station and depth, 50 L of seawater were collected in 8-10 L Niskin bottles, mounted on a General Oceanics 12-bottle rosette deployed from the vessels. To acquire enough water for the samples described herein, in addition to other biological and physico-chemical samples as mentioned above, usually two casts were made per station: one from each of the epi- and mesopelagic zones. The samples were size fractionated. All equipment for filtration and size-fractionation was rinsed twice with dH$_2$O between each sample. During the January and March cruises, the samples were prefiltered through a 180 $\mu$m mesh size nylon filter, and size fractionated into the 3-180 $\mu$m and 0.45-3 $\mu$m fractions by filtration using a peristaltic pump (Masterflex 07523-80, ColeParmer, IL, USA), through serially connected 3 $\mu$m and 0.45 $\mu$m polycarbonate filters (Isopore/Durapore, 142 mm diameter, Millipore, Billerica, MA, USA), mounted in stainless steel tripods (Millipore). The filters were removed from the filter holders and cut in four. Two of the pieces were used for DNA extraction, the others were saved for other purposes. The pieces for DNA were transferred to a 50 mL Falcon tube with 1 mL (65 °C) AP1 lysis buffer (Qiagen, Hilden, Germany), the plankton material was washed off the filters, and buffer with material and the filters were transferred to two separate cryovials. AP1 buffer (65 °C) was added to the vial with the filters, flash frozen in liquid nitrogen and kept at -80 °C until DNA extraction. During the May, August and November cruises the water was sequentially filtered through 200, 50, and 10 $\mu$m nylon mesh, the material on each nylon mesh was collected with sterile filtered seawater in a 50 mL Falcon tube, and collected by filtration on a polycarbonate filter (10 $\mu$m pore size 47 mm diameter, Millipore). The filters were transferred to cryovials to which 1 mL of warm AP1 buffer was added, flash frozen in liquid nitrogen and kept at -80 °C until DNA extraction. The size fraction < 10 $\mu$m passing through the nylon mesh system was fractionated into the 3-10 $\mu$m and 0.45-3 $\mu$m size fractions by serial filtration through 142 mm diameter polycarbonate filters as described above.

### 3.1.2 Net hauls

Vertical phytoplankton net hauls (mesh size 10 $\mu$m) were collected between 50 m depth and the surface at each station in May, August and November. The net haul samples were diluted to 1 L with sterile filtered sea water, and size fractionated by filtration through 200, 50 and 10 $\mu$m nylon mesh. The plankton was washed off the nylon mesh with sterile sea water, diluted to 50 mL in a Falcon tube and a 20 mL aliquot collected on a 10 $\mu$m pore size polycarbonate filter and preserved for DNA extraction as described above. The remaining 30 mL were preserved for microscopical analyses to be reported separately. An overview over which type of samples and size fractions that are available from each cruise can be found in Table 2.

### 3.2 DNA extraction

DNA was extracted with the DNeasy Plant mini kit (Qiagen), according to the protocol from the manufacturer, except for the following step: To disrupt the thick cell walls of certain protist groups, the frozen samples in cryovials were incubated at 95 °C for 15 min, then shaken in a bead-beater 2x 45-60 s. Subsequently, 4 $\mu$L RNase was added, and the lysate was incubated on a heating block at 65 °C for 15-20 min, with vortexing in-between. Purity and quantity of the extracted DNA was assessed with NanoDrop.

## 4 18S rRNA gene amplicon generation for eukaryotic metabarcoding

### 4.1 PCR amplification and Illumina sequencing

The V4 region of the 18S rRNA gene was amplified with the primer pair 18S TAReuk454FWD1 (5'-CCAGCASCYGCGGTAATTCC-3') and V4 18S Next.Rev (5'-ACTTTCGTTCTTGATYRATGA-3') (Piredda et al., 2017), which yields a fragment of 410-420 base pairs. This primer pair is an improvement over the widely used primers developed by (Stoeck et al., 2010) which have been used in more than 60 studies. The Piredda primers aim of reducing the biases against Haptophyta seen in the Stoeck primers (Piredda et al., 2017). The samples were prepared for Illumina sequencing with a so-called dual-index approach (e.g. Fadrosh et al., 2014), where a 12 bp internal barcode was added to both the forward and reverse amplification primers for the initial amplification. In order to pool several samples into one library preparation, 19 unique barcodes for each direction were used. The internal barcodes were designed to give a balanced distribution of the four bases, following the recommendations of Fadrosh et al. (2014). PCR reactions consisted of 12.5 $\mu$L KAPA HiFi HotStart ReadyMix 2x (KAPA Biosystems, Wilmington, MA, USA), 5 $\mu$L of each primer (1 $\mu$M), 10 ng DNA template and PCR-grade water to a final volume of 25 $\mu$L. The PCR was run on an Eppendorf thermocycler (Mastercycler, ep gradient S, Eppendorf), with an initial denaturation step at 95 °C for 3 min, followed by 25 cycles of denaturation at 98 °C for 20 s, annealing at 65 °C for 60 s and elongation at 72 °C for 1.5 min, and a final elongation step at 72 °C for 5 min. The reactions were performed in triplicate for each sample and pooled prior to purification and quantification. The length of the PCR products was assessed by gel electrophoresis. In all samples, there was a strong band at about 470 bp, and no other bands (data not shown). The PCR products were purified with AMPure XP beads (Beckman Coulter, Brea, USA) using the standard protocol with elution buffer EB (Qiagen), quantified with a Qubit dsDNA

High-Sensitivity kit (Thermo Fisher, Waltham, MA, USA) and pooled in equal concentrations to create nine pools with ca. 19 samples in each. The pools were sent to library preparation at the Norwegian Sequencing Centre (Oslo, Norway) and GATC GmbH (Konstanz, Germany) with the KAPA library amplification kit (Kapa Biosystems). Further quality control of the amplicons were made with Bioanalyzer at the sequencing centres prior to Illumina sequencing. Due to world-wide supply problems with the Illumina MiSeq chemistry in 2015, the sequencing was done with a modified HiSeq protocol on two HiSeq runs at the GATC Centre in October 2015. This modified protocol yielded 250 bp paired-end reads. The HiSeq sequencing runs were spiked with 20 % PhiX (viral DNA added to ensure homogeneity of bases during sequencing). To assess variation between DNA extracts and annealing temperature, we sequenced separately replicate DNA extractions and replicate PCR runs with 60 °C annealing temperature for a few samples (indicated in Table 3). After initial analysis of the HiSeq data, samples with low number of reads were re-amplified with 30 cycles with the original DNA as template to increase the concentration of the PCR product, and re-sequenced with Illumina MiSeq at the Norwegian Sequencing Centre. The MiSeq protocol yielded 300 bp paired-end reads. In total, we sequenced 199 samples separately. The taxonomic composition of sequencing replicates of the same sample was inspected visually, and in all cases the different sequencing protocols were found to give similar taxonomic compositions.

## 4.2   Bioinformatics processing

PhiX sequences were removed and the raw reads were sorted according to the Illumina index by the Illumina software at the sequencing provider. For the HiSeq datasets, the samples within each Illumina library were demultiplexed with cutadapt v2.10 with Python 3.6.11 (Martin, 2011), requiring 0 errors in the internal barcodes. The amplification primers were removed with cutadapt v2.8 with Python 3.7.6, with setting --trim-n (trim N's on ends of reads). The reads were denoised and merged with dada2, v1.16. (Callahan et al., 2016). For the HiSeq reads the settings were: truncLen = c(240,200), minLen = c(240,200), truncQ = 2, maxEE = c(10, 10), max_number_asvs = 0. Chimeras were detected with isBimeraDenovo with default settings, and removed with removeBimeraDenovo, with 'method_chimera' = "pooled". For the MiSeq reads truncLen and minLen were set to c(270, 240), the other settings were the same as for HiSeq. The reads were subsequently classified with assignTaxonomy, the dada2 implementation of the naive Bayesian classifier method (Wang et al., 2007), against the Protist Ribosomal Reference Database, version 4.12.0 (Guillou et al., 2013, https://github.com/pr2database/pr2database/releases/tag/v4.12.0). ASVs with less than 90 % bootstrap value at class level and/or which comprised less than 10 reads in total were removed. As this study is focusing on the protists, all reads assigned to Metazoa and Viridiplantae (Embryophyceae) were excluded from the processed ASV tables (Table 4).

## 4.3   Preparation of ASV-tables

Preparation of the ASV-tables was done in R v. 3.6.0 (R Core Team, 2019). To be able to compare ASV richness between samples (i. e. the number of ASVs in each sample), the DNA samples were subsampled to equal read number. Prior subsampling, data from fastq files that map onto the same size-fractionated DNA sample were merged by taking the sum of the read number for each ASV. The merging was done to increase the number of reads in the samples that initially had a low read number

after HiSeq sequencing. The DNA samples were subsampled according to size fraction as follows: 0.45-3 $\mu$m: 40,000 reads, 3-180 $\mu$m: 88,000 reads, 3-10 $\mu$m: 40,000 reads, 10-50 $\mu$m: 40,000 reads, 50-200 $\mu$m: 8,000 reads. It should be noted that the number of reads in the 3-180 $\mu$m fraction samples was in some cases lower than the subsample size. Subsampling to equal read number was performed 100 times, and the average read number per ASV was used, rounded to 0 decimals. Subsampling was done with the function rrarefy() from the 'vegan' package (Oksanen et al., 2020), v. 2.5-7. The low number of protist reads in the 50-200 $\mu$m fraction was due to a high proportion of Metazoan reads in this fraction. To calculate the ASV richness of each sample, the subsampled ASV-table was transformed to presence-absence. An overview of the available versions of the ASV-table is given in Table 4. To assess whether the sequencing depth was sufficient, we plotted rarefaction curves for each DNA sample, and calculated the slope at the endpoint of the curve with the functions 'rarecurve' and 'rareslope', respectively. Figures were made with the R package 'ggplot2' (Wickham, 2016). In addition to the figures and tables presented in the paper, interactive versions of figures, tables, and supplementary material are available as a Shiny app (Chang et al., 2019) (available at https://micropolar-protists.metapr2.org/). Interactive figures and tables were made with the packages 'DT' (Xie et al., 2020) and 'plotly' (Sievert, 2020), respectively.

## 5 Data description

### 5.1 Overview of sequenced samples

In total we obtained 44 water samples from Niskin bottles and 8 net hauls, which were fractionated into 140 and 15 size-fractionated samples, respectively (the DNA isolation of the 10-50 $\mu$m fraction from the net haul taken at station P04 in May failed). The samples from Niskin and net hauls are in the following referred to as DNA samples, and denoted month_station_depth_minfract_maxfract or month_station_net_minfract_maxfract, respectively. On some DNA samples, we performed replications of DNA extraction, PCR with variable annealing temperature, and/or replicate sequencing. Thus, one or more fastq-file pairs can map onto the same DNA sample. The fastq-files were deposited individually to the European Nucleotide Archive (ENA, project accession number PRJEB40133), and are referred to as a 'sequencing event' ('seq_event') in Table 3. In total the dataset consists of 199 sequencing events, some of which were merged, to form in total 155 DNA samples. Description of metadata available for each fastq-file pair can be found in in Table 3. Table 1 describes all environmental parameters obtained from each water acquisition event (i.e. from the Niskin samples). These are referred to as 'env_sample' and labelled month_station_depth.

### 5.2 Total number of reads and ASVs

After quality filtering, dada2 processing, removal of chimeras, and non-target taxonomic groups, the dataset comprised 6,536 protist ASVs, corresponding to 32,164,445 reads. After subsampling to equal number of reads per sample within each size fraction, the data set was reduced to 6,430 ASVs and 5,729,358 reads. Number of ASVs per division or class within each size fraction, after subsampling, is shown in Table A1. In total, we recovered 3,339; 2,720; 2,799; 1,153 and 3,172 ASVs in the

0.45-3, 3-10, 10-50, 50-200 and 3-180 $\mu$m size fractions, respectively. Note that the numbers are not directly comparable, as the fractions were not obtained from the same number of samples (e.g. 3-180 $\mu$m were only sampled in January and March). Syndiniales and Dinophyceae had the highest number of assigned ASVs, with 2,166 and 1,723, respectively. Ciliophora, Bacillariophyta, Radiolaria and Chlorophyta had between 400 and 200 assigned ASVs each (Table A1).

### 5.3 Sample saturation

Slopes of rarefaction curves at the endpoint, after subsampling, ranged from 0 to 0.014 (Figure 3) which means that for every 1000 extra reads sequenced, we could expect to find between 0 and 14 new ASVs (de Vargas et al., 2015). There was no correlation between the number of ASVs detected in a sample and the slope of the rarefaction curve ($r^2$ = -0.13, p = 0.11), which means that the DNA samples with a low number of ASVs were not necessarily under-sampled.

### 5.4 Variation in taxonomic composition by season, depth and size fraction.

The proportional taxonomic composition of the metabarcoding reads, at division or class level, is shown in Figure 4. The taxonomic composition of the ASV richness in each sample is shown in Figure A1. The metabarcoding data reveal variation in taxonomic composition both by season and depth, in all size fractions. In the following, the fractions are defined as follows: 0.45-3 $\mu$m = picoplankton, 3-180 $\mu$m = nano-micro, 3-10 $\mu$m = small nanoplankton, 10-50 $\mu$m = large nanoplankton and 50-200 $\mu$m = microplankton. All the major protist groups varied from less than 1 % of the reads, to up to 99 % for the most abundant (e.g. Syndiniales in the picoplankton fraction, and diatoms in the microplankton fractions; Table A2).

In January (winter) and March (pre-bloom), heterotrophic or parasitic groups (e. g. certain dinoflagellates, Syndiniales and Picozoa) were dominating at all depths. In the picoplankton size fraction, the parasitic dinoflagellate group Syndiniales had the highest relative abundance these months, with up to 99 % of the reads, followed by the heterotrophic group Picozoa, with up to 35 % of the reads, and Pseudofungi with up to 12 % (previously categorised as Marine Stramenopiles, MAST). Syndiniales also had the highest ASV richness in all samples. In the nano-micro fraction, Dinophyceae had generally higher relative abundance, with 20-55 % of the reads in most samples. Syndiniales and Picozoa had up to 82 % and 40 %, respectively. Syndiniales had highest ASV richness also in this fraction, followed by Dinophyceae. Other heterotrophic groups notably present in this fraction were Pseudofungi and Radiolaria, with 2-20 % of the reads each, and Ciliophora and Choanoflagellida with up to 6 % of the reads. ASVs assigned to phototrophic groups (e. g. diatoms, haptophytes and chlorophytes) were detected in these months, but constituted less than 3 % of the reads in all samples.

The May samples were characterised by higher proportions of phototrophs in all size fractions. In the pico- and small nanoplankton fractions, there was a pronounced difference between the epipelagic and mesopelagic samples this month. In the picoplankton fraction, Chlorophyta (mainly represented by the genera *Micromonas* and *Bathycoccus*) had high relative abundance in the epipelagic samples, with 17-43 % of the reads. In the small nanoplankton fraction, Haptophyta (mainly represented by the genus *Phaeocystis*) and Dinophyceae were the most abundant groups in the euphotic samples with 25-47 % and 14-39 % of the reads, respectively. The mesopelagic samples in the pico- and small nanoplankton fractions were characterised by high abundance of Syndiniales, with 47-85 % of the reads. In these fractions, ASV richness was generally

higher in the mesopelagic than in the epipelagic samples. Syndiniales generally had the highest number of ASVs, despite having lower relative abundance. In the large nanoplankton and microplankton fractions, diatoms were dominating both in the epi- and mesopelagic samples, with up to 99 % of the reads. Dinoflagellates (Dinophyceae) was the second most abundant group in the large nanoplankton fraction, with up to 50 % of the reads. In the microplankton fraction, *Phaeocystis* was also abundant in certain samples, with up to ca. 30 % of the reads. In the net haul samples from May, the diatoms were dominating

with up to 97 % of the reads. Dinophyceae had 10-11 %, and Haptophyta constituted 11 % in the microplankton fraction from station P01. These fractions generally had lower ASV richness than the pico-nano, and there was no clear difference in ASV richness by depth. The groups with highest ASV richness in these samples were Dinophyceae, Bacillariophyta and Syndiniales.

In August, in the picoplankton fraction of the epipelagic samples, Dinophyceae had the highest relative abundance, with 13-64 %. Haptophyta had ca. 4-14 % and Chlorophyta ca. 7-25 % in these samples. In the mesopelagic samples in this fraction,

Syndiniales also dominated in August, with up to 68 % of the reads. Radiolaria accounted for 10-13 % of the reads in these samples, whereas Picozoa had 3-15 %. Picozoa relative abundance reached also up to 12 % in the epipelagic samples. In the nanoplankton size fraction, Dinophyceae was dominating, with 27-79 % of the reads. Similar to in May, Syndiniales had generally highest ASV richness in the picoplankton fraction. In the nanoplankton fraction, Syndiniales and Dinophyceae had similar ASV richness. In the large nanoplankton fraction, Dinophyceae dominated with 31-88 % of the reads. Diatoms constituted up

to 41 % of the reads, and there was no clear difference in proportion of this group between the epi- and mesopelagic samples. In the microplankton fraction, the diatoms dominated, with 30-73 % of the reads. Dinophyceae was the second most abundant group in this fraction, with 3-46 % of the reads. In the net haul samples, Dinophyceae and diatoms were the most abundant in the large nanoplankton and microplankton fraction, with 64-77 % and 12-20 % of the reads, respectively. In the microplankton fraction Radiolaria were also abundant, representing 7-30 % of the reads. ASV richness was slightly higher than in May in the

large nanoplankton and microplankton fraction. Syndiniales and Dinophyceae had the highest richness also in these fractions, followed by diatoms and ciliates.

In November, the proportion of reads assigned to phototrophs was less than 3 % in most samples in the pico- and nanoplankton. In the large nanoplankton and microplankton fraction, diatom reads constituted 1-33 %. In the pico fraction, Syndiniales and Picozoa were the most abundant, with 40-75 % and up to 25 % of the reads, respectively. Radiolaria represented 44 %

of the reads in the sample N04_1000. Dinophyceae was the most abundant group in the fractions between 3 and 50$\mu$m, with 28-76 % of the reads. In the microplankton fraction, Radiolaria, Dinophyceae and Syndiniales were the most abundant with up to 77, 43 and 42 % of the reads, respectively. In the net hauls, Ciliophora was also abundant, with up to ca. 30 % of the reads in each size fraction. ASV richness was generally higher this month than in May and August, especially in the small and large nanoplankton fractions fractions. Syndiniales and Dinophyceae had highest richness also this month.

**6 Conclusions**

This dataset offers novel insights into the spatial and seasonal diversity and taxonomic composition of the protist community in the Atlantic gateway to the Arctic Ocean. It is the first study to provide data on the eukaryote microbial community throughout

a complete year and down to 1000 m in this area of the Arctic. It forms the basis for future studies to detect changes in the eukaryote microbial community, and for more detailed studies on the dynamics and community structure of specific taxonomic groups.

*Code and data availability.* The fastq files with raw 18s rRNA V4 reads are available on the European Nucleotide Archive repository under project number PRJEB40133. The untransformed ASV table, meta data table and a table with environmental data are deposited in the Sea scientific open data publication repository (SEANOE), under the CC-BY license, with doi: https://doi.org/10.17882/79823/, last access: 19 April 2021 (Egge et al., 2014). The ASV tables, including the ASV sequences and assigned taxonomy, R-scripts for producing the figures and tables, and a Shiny application with interactive versions of the figures and tables are deposited on GitHub: https://github.com/EEgge/micropolar_protists_datapaper. The Shiny app can be found at https://micropolar-protists.metapr2.org/, or opened in RStudio by running the following command: shiny::runGitHub("micropolar_protists_datapaper","EEgge", ref = "main").

*Author contributions.* Conceptualization: AL, GB, BE, DV, UJ, Data curation: EE, BE, DV, Formal analysis: EE, BE, DV, Funding acquisition: AL, GB, BE, DV, UJ, Investigation: AL, GB, BE, UJ, SE, EE, Project administration: AL, GB; Visualization: EE, DV, BE; Writing - original draft preparation: EE, BE, DV; Writing - review and editing: EE, BE, DV, AL, GB, SE, UJ. All authors read and approved the final version of the paper.

*Competing interests.* No competing interests are present.

*Acknowledgements.* The study was conducted as part of the Research Council of Norway supported project Micropolar - Processes and Players in Arctic Marine Pelagic Food Webs - Biogeochemistry, Environment and Climate Change no. 225956/E10 (prosjektbanken.forskningsradet.no/#/projec DV was supported by ANR contract PhytoPol (ANR-15-CE02-0007). We wish to thank members of the MicroPolar and Carbon Bridge projects for assisting in the sampling campaigns, and the crews at K/V Svalbard (January cruise), R/V Lance (March) and R/V Helmer Hanssen (May, August and November cruise).

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

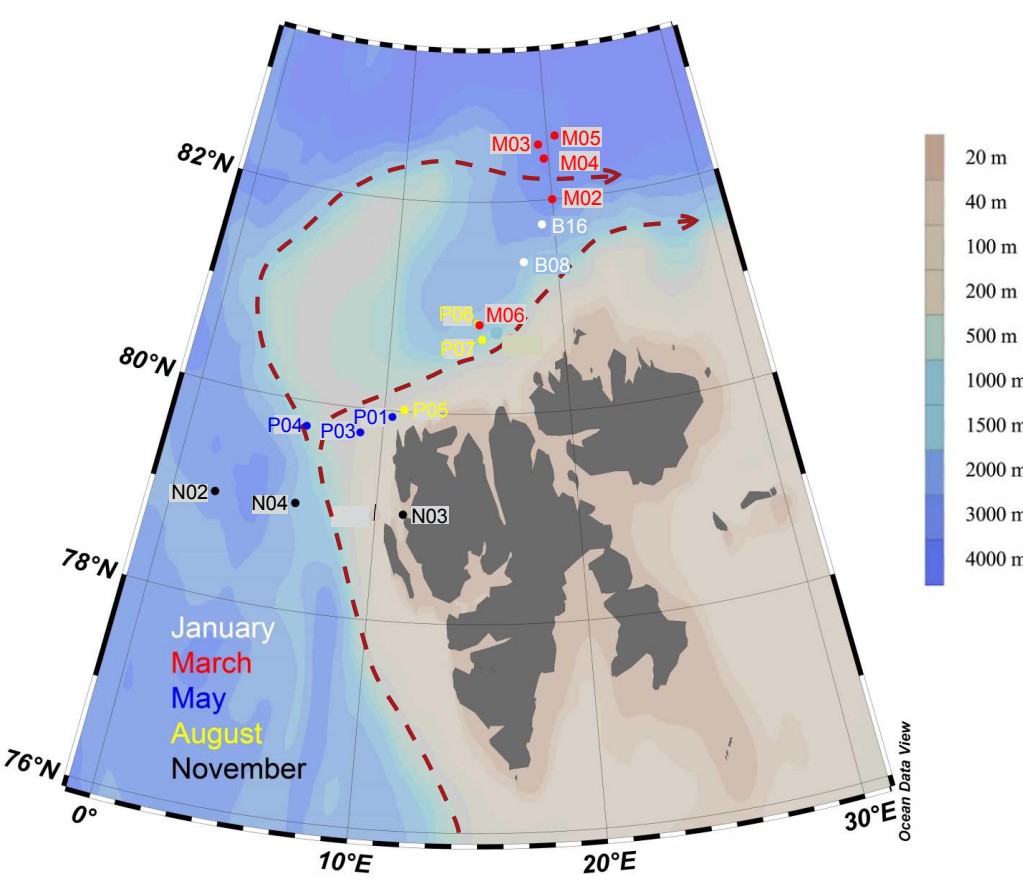

**Figure 1.** Map of sampling locations of the MicroPolar sampling campaign. Color correspond to cruise month. The red dashed line indicates the major flow patterns of warm Atlantic Water into the Arctic Ocean. Color scale bar indicates bottom depth. Red arrows indicate the main flow of the West Spitsbergen Current, according to Cokelet et al. (2008) and Randelhoff et al. (2018).

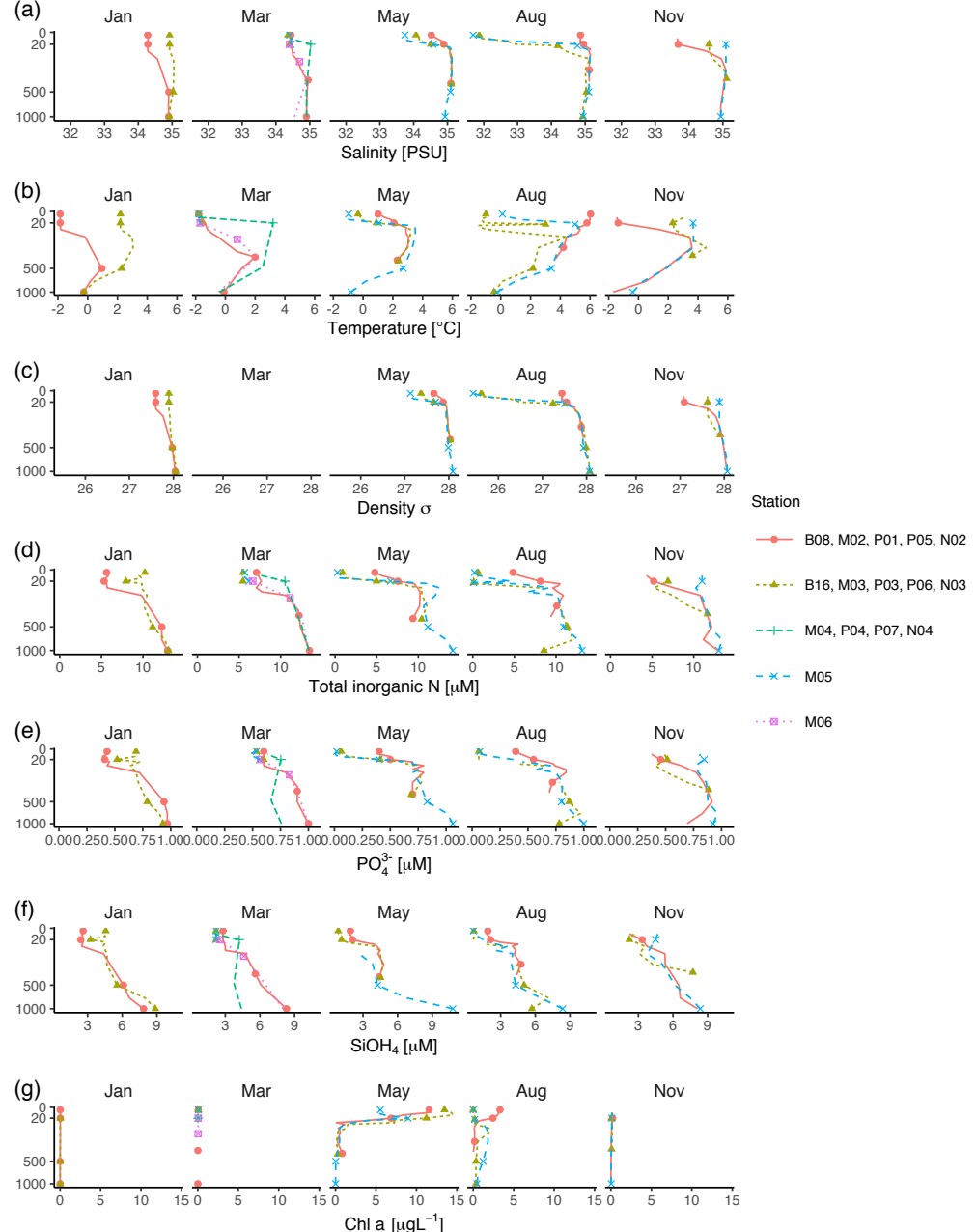

**Figure 2.** Profiles of environmental variables measured during the MicroPolar cruises. (a) Salinity [PSU], (b) Temperature [°C], (c) Density [$\sigma$], (d) Total inorganic N [$\mu$M], (e) $PO_4^{3-}$ [$\mu$M] (f) $SiOH_4^-$ [$\mu$M], (g) Chl *a* [$\mu$gL$^{-1}$]. Data obtained from (Paulsen et al., 2017). Points indicate the depths where samples for protist metabarcoding were taken with Niskin bottles. To better distinguish between data points in the epipelagic zone, the y-axis is square root-transformed. The full profile was not available from all stations.

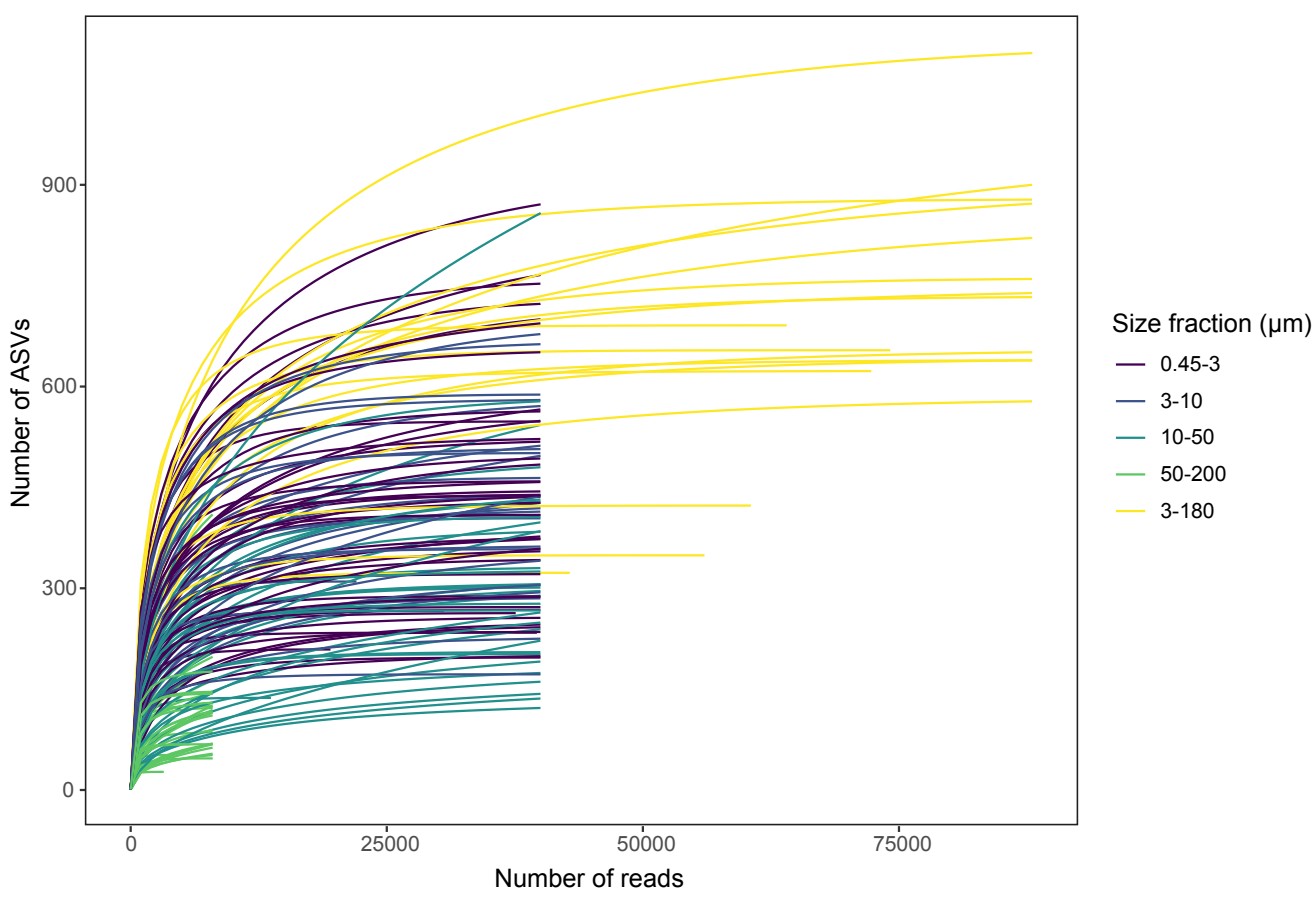

**Figure 3.** Rarefaction curves for each DNA sample, after subsampling to equal number of reads per sample within each size fraction. Unequal number of reads per sample in the 3-180 $\mu$m fraction was due to low sequencing output in some of these samples.

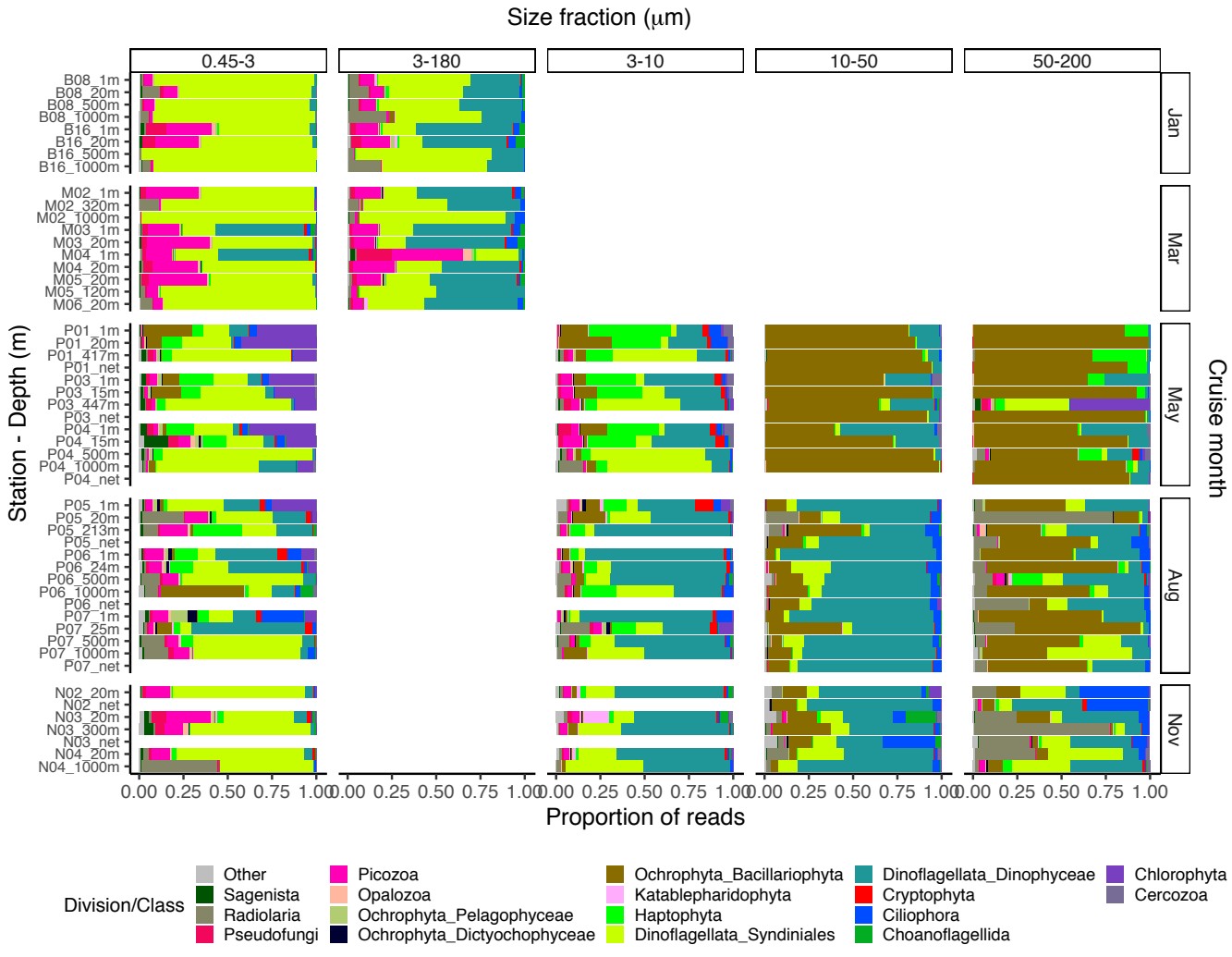

**Figure 4.** Barplot of relative read abundance of the major protist divisions or classes in each size-fractionated sample. Size fraction 3-180/3-10 corresponds to 3-180 $\mu$m in January and March, and 3-10 $\mu$m otherwise. The 3-180 $\mu$m fraction was only sampled in January and March. In May, August and November this fraction was replaced by the 3-10, 10-50 and 50-200 $\mu$m fractions. Net hauls were sampled in May, August and November, and were fractionated into the 10-50 and 50-200 $\mu$m fractions. Diatoms are denoted as "Ochrophyta_Bacillariophyta" in the legend.

**Table 1.** Overview of the environmental data available from the water samples corresponding to the metabarcoding DNA samples presented in this paper, listed in the file "env_data_depths.txt" (see "Code and data availability" section). The original data set can be found in Paulsen et al. (2017).

| Category | Variable names and units |
|---|---|
| Sample name | env_sample (month_station_depth) |
| Time | date_dd.mm.yy, month |
| Light conditions | daylength [h], euphotic zone depth [m] in May and August |
| Station name | station |
| Location | latitude (N), longitude (E) |
| Depth | bottom depth, sampling depth [m] |
| Physical | temperature [°C], salinity [PSU], density $\sigma$-t [kg/m$^3$], oxygen [$\mu$mol/l], oxygen saturation [%] |
| | pressure [dbar], turbidity [Nephelometric turbidity unit, NTU] |
| Inorganic nutrients | $NH_4^+$, $NO_2^-$, $NO_2^- + NO_3^-$, $PO_4^-$, $SiOH_4^-$, total inorganic N [$\mu$mol/l] |
| Organic compounds (Dissolved, Particulate, Total) | carbon, nitrogen [$\mu$mol/l] |
| Chlorophyll | total Chl $a$, Chl $a$ < 10 $\mu$m [$\mu$gL$^{-1}$] |
| Fluorescense | fluorescense [RFU] |
| Counts | virus (small, medium, large, total [mL$^{-1}$]), |
| | heterotrophic bacteria [mL$^{-1}$], *Synechococcus* [mL$^{-1}$], |
| | picophytoplankton [mL$^{-1}$], nanophytoplankton [L$^{-1}$], |
| | heterotrophic nanoflagellates (HNF) [mL$^{-1}$] |

**Table 2.** Overview of which stations, depths, type of samples and size fractions were sampled each of the cruise months.

| Cruise month | Station | Depths (m) | Type of samples | Size fractions ($\mu$m) |
|---|---|---|---|---|
| January | B08 | 1, 20, 500, 1000 | Niskin | 0.45-3, 3-180 |
| January | B16 | 1, 20, 500, 1000 | Niskin | 0.45-3, 3-180 |
| March | M02 | 1,320, 1000 | Niskin | 0.45-3, 3-180 |
| March | M03 | 1, 20 | Niskin | 0.45-3, 3-180 |
| March | M04 | 1, 20 | Niskin | 0.45-3, 3-180 |
| March | M05 | 20, 120 | Niskin | 0.45-3, 3-180 |
| March | M06 | 20 | Niskin | 0.45-3, 3-180 |
| May | P01 | 1, 20, 417 | Niskin and Net haul | 0.45-3, 3-10, 10-50, 50-200 |
| May | P03 | 1, 15, 447 | Niskin and Net haul | 0.45-3, 3-10, 10-50, 50-200 |
| May | P04 | 1, 15, 500, 1000 | Niskin and Net haul | 0.45-3, 3-10, 10-50, 50-200 |
| August | P05 | 1, 20, 213 | Niskin and Net haul | 0.45-3, 3-10, 10-50, 50-200 |
| August | P06 | 1, 24, 500, 1000 | Niskin and Net haul | 0.45-3, 3-10, 10-50, 50-200 |
| August | P07 | 1, 25, 500, 1000 | Niskin and Net haul | 0.45-3, 3-10, 10-50, 50-200 |
| November | N02 | 20 | Niskin and Net haul | 0.45-3, 3-10, 10-50, 50-200 |
| November | N03 | 20, 300 | Niskin and Net haul | 0.45-3, 3-10, 10-50, 50-200 |
| November | N04 | 20, 1000 | Niskin and Net haul | 0.45-3, 3-10, 10-50, 50-200 |

**Table 3.** Description of metadata table (named "meta_data_fastqfiles.txt") for the fastq files deposited in ENA. These metadata can be joined with environmental data (described in Table 1) by the 'env_sample' column. Each fastq file is unique, but two or more fastq files may map onto the same DNA-extract and/or PCR.

| Column name | Description |
| --- | --- |
| filename | Name of fastq file |
| seq_event | Sample name including barcode and library numbers |
| accno | Accession number European Nucleotide Archive |
| env_sample | Code for water sample (format: month_station_depth) |
| sample_sizefract | Code for size-fractionated sample (format: month_station_depth_minfract_maxfract) |
| fraction_min | Lower limit of size fraction |
| fraction_max | Higher limit of size fraction |
| coll_method | Collection method (Niskin bottle or net haul) |
| dna_concentration | DNA concentration (ng/$\mu$L) |
| 260_280 | Ratio A260 over A280 of isolated DNA |
| 260_230 | Ratio A260 over A230 of isolated DNA |
| seq_method | Sequencing method (Illumina HiSeq or MiSeq) |
| pcr_cycles | Number of PCR cycles |
| n_reads | Number of reads after processing with cutadapt (as described in Methods) |
| comment | Comments regarding replicate DNA extraction, PCR annealing temp. and/or replicate sequencing |

**Table 4.** Overview over ASV-tables. Commands for creating ASV tables 1-3 from the original ASV table are found in the script 'asvtables.R'. From ASV tables 1-3, ASVs assigned to division Metazoa or class Embryophyceae have been removed. ASV tables 1-3 are also available as proportions and presence-absence. The original ASV-table produced after dada2 processing contain one 'sequencing_event' less than the number of fastq files deposited in ENA, due to low quality of reads in this particular file.

| Name | Description |
| --- | --- |
| metapr2_wide_asv_set_207_208_209_Eukaryota.xlsx | Original ASV table after processing with dada2, including taxonomic classification against PR2. |
| asvtab1_nonmerged_readnum.txt | 'Sequencing events' (i.e. sequencing replicates of DNA samples) kept separate, not subsampled. ASVs assigned to Metazoa and Embryophyceae removed. |
| asvtab2_merged_readnum.txt | Sequencing replicates of DNA samples merged |
| asvtab3_merged_subsamp_readnum.txt | Sequencing replicates of DNA samples merged, then all DNA samples are subsampled to equal read number within each size fraction. |

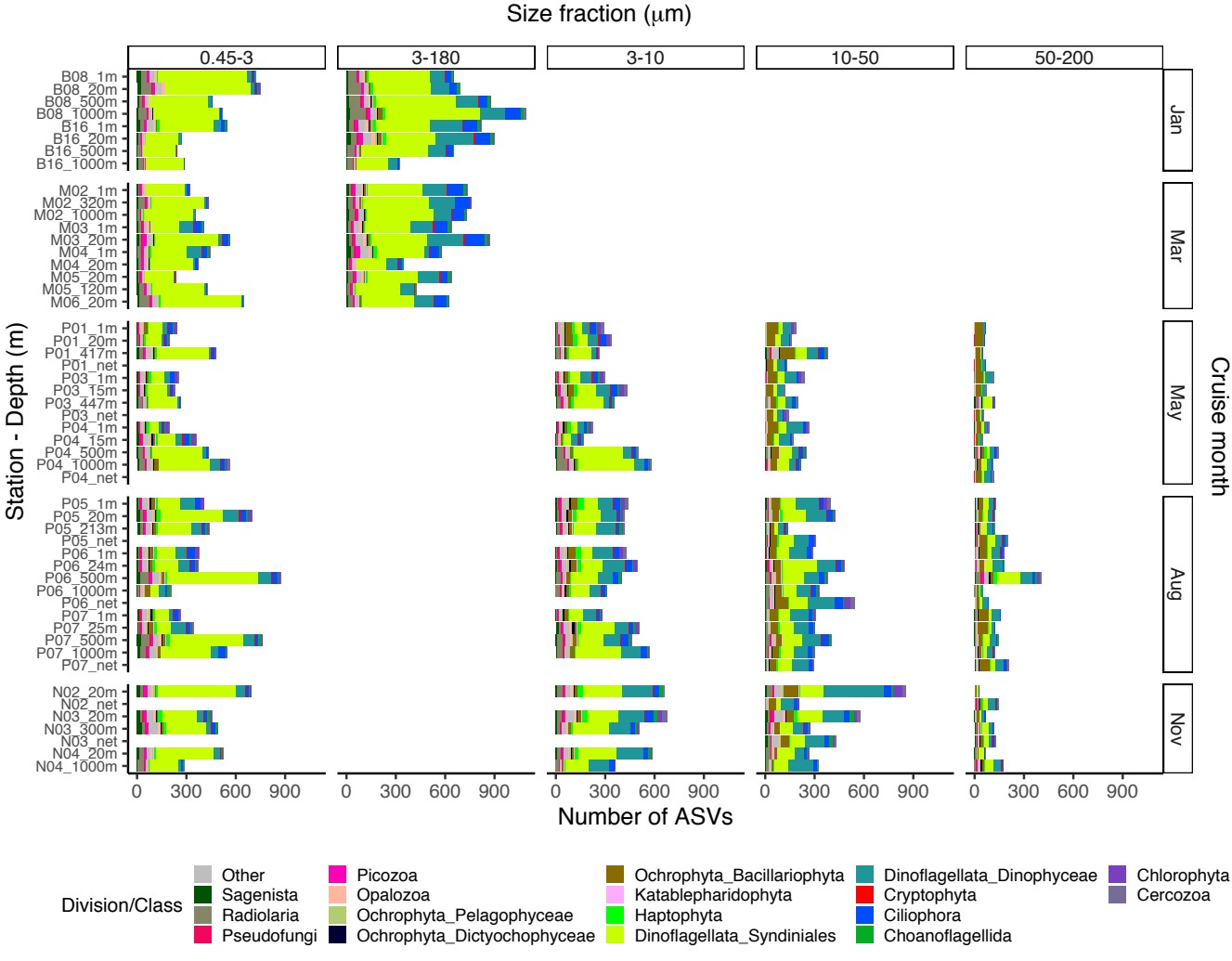

**Figure A1.** Barplot of the taxonomic composition at the division or class level of the ASV richness in each DNA sample, based on the ASV table subsampled to equal read number per sample within each size fraction. The 3-180 $\mu$m fraction was only sampled in January and March. In May, August and November this fraction was replaced by the 3-10, 10-50 and 50-200 $\mu$m fractions. Net hauls were sampled in May, August and November, and were fractionated into the 10-50 and 50-200 $\mu$m fractions.

405 **Appendix A: Supplementary material**

Table A1: Number of ASVs assigned to each division or class, distributed by size fraction, and in total. Note that a given ASV may occur in multiple size fractions. Number of samples in each size fraction: 0.45-3$\mu$m: 44, 3-180$\mu$m: 18, 3-10$\mu$m: 26, 10-50$\mu$m: 33, 50-200$\mu$m: 34.

| Division/Class | Size fraction ($\mu$m) | | | | | |
| --- | --- | --- | --- | --- | --- | --- |
| | 0.45-3 | 3-10 | 10-50 | 50-200 | 3-180 | Total |
| Total | 3339 | 2720 | 2799 | 1153 | 3172 | 6536 |
| Dinoflagellata_Syndiniales | 1741 | 1061 | 638 | 335 | 1377 | 2166 |
| Dinoflagellata_Dinophyceae | 318 | 553 | 975 | 210 | 636 | 1723 |
| Ciliophora | 201 | 207 | 174 | 57 | 333 | 454 |
| Ochrophyta_Bacillariophyta | 69 | 94 | 286 | 190 | 61 | 415 |
| Radiolaria | 189 | 136 | 104 | 69 | 199 | 275 |
| Chlorophyta | 111 | 99 | 105 | 34 | 30 | 206 |
| Cercozoa | 63 | 100 | 99 | 56 | 31 | 177 |
| Haptophyta | 100 | 103 | 44 | 38 | 57 | 172 |
| Sagenista | 70 | 30 | 50 | 20 | 55 | 119 |
| Opalozoa | 63 | 41 | 40 | 15 | 49 | 103 |
| Picozoa | 66 | 32 | 19 | 14 | 58 | 92 |
| Ochrophyta_Chrysophyceae | 59 | 33 | 29 | 9 | 36 | 89 |
| Telonemia | 32 | 42 | 16 | 11 | 52 | 70 |
| Choanoflagellida | 30 | 32 | 37 | 12 | 36 | 63 |
| Fungi | 16 | 15 | 30 | 11 | 21 | 54 |
| Pseudofungi | 24 | 14 | 25 | 13 | 23 | 46 |
| Ochrophyta_Bolidophyceae | 29 | 11 | 12 | 9 | 17 | 41 |
| Cryptophyta | 27 | 14 | 6 | 7 | 14 | 35 |
| Ochrophyta_Pelagophyceae | 18 | 21 | 12 | 11 | 16 | 33 |
| Ochrophyta_Dictyochophyceae | 26 | 24 | 17 | 9 | 12 | 32 |
| Stramenopiles_X | 26 | 9 | 3 | 3 | 14 | 29 |
| Centroheliozoa | 6 | 8 | 23 | 6 | 9 | 28 |
| Apicomplexa | 4 | 5 | 19 | 2 | 3 | 21 |
| Katablepharidophyta | 6 | 7 | 3 | 2 | 6 | 11 |
| Alveolata_X | 8 | 3 | 1 | 1 | 5 | 10 |
| Ochrophyta_MOCH-1 | 7 | 4 | 0 | 1 | 5 | 10 |
| Ochrophyta_MOCH-2 | 5 | 6 | 3 | 1 | 4 | 10 |

| Division/Class | Size fraction ($\mu$m) | | | | | Total |
|---|---|---|---|---|---|---|
| | 0.45-3 | 3-10 | 10-50 | 50-200 | 3-180 | |
| Mesomycetozoa | 3 | 3 | 6 | 2 | 0 | 8 |
| Dinoflagellata_Dinophyta_X | 7 | 3 | 1 | 1 | 6 | 7 |
| Ochrophyta_Phaeophyceae | 3 | 3 | 6 | 2 | 0 | 7 |
| Perkinsea | 5 | 2 | 0 | 0 | 0 | 6 |
| Rhodophyta | 1 | 2 | 4 | 1 | 0 | 5 |
| Dinoflagellata_Noctilucophyceae | 1 | 0 | 4 | 0 | 2 | 4 |
| Opisthokonta_X | 1 | 1 | 2 | 1 | 2 | 3 |
| Streptophyta | 1 | 0 | 1 | 0 | 1 | 3 |
| Lobosa | 0 | 0 | 2 | 0 | 0 | 2 |
| Apusomonadidae | 0 | 0 | 0 | 0 | 1 | 1 |
| Conosa | 0 | 0 | 1 | 0 | 0 | 1 |
| Dinoflagellata_Ellobiophyceae | 1 | 1 | 0 | 0 | 1 | 1 |
| Discoba | 1 | 0 | 0 | 0 | 0 | 1 |
| Metamonada | 0 | 0 | 1 | 0 | 0 | 1 |
| Ochrophyta_MOCH-3 | 0 | 1 | 1 | 0 | 0 | 1 |
| Ochrophyta_MOCH-4 | 1 | 0 | 0 | 0 | 0 | 1 |

Table A2: Minimum and maximum percentage of reads of each division or class in each size fraction. The entries have the format 'min %, max' %.

| Division/Class | Size fraction (µm) | | | | |
|---|---|---|---|---|---|
| | 0.45-3 | 3-10 | 10-50 | 50-200 | 3-180 |
| Dinoflagellata_Syndiniales | 6.4e+00, 9.9e+01 | 3.0e+00, 6.4e+01 | 3.2e-01, 2.6e+01 | 1.3e-02, 4.8e+01 | 1.3e+01, 8.2e+01 |
| Dinoflagellata_Dinophyceae | 3.0e-02, 6.4e+01 | 9.9e+00, 7.9e+01 | 1.0e+00, 8.8e+01 | 1.0e-01, 4.6e+01 | 1.8e+00, 5.6e+01 |
| Ciliophora | 0.0e+00, 2.4e+01 | 3.5e-01, 8.9e+00 | 2.5e-03, 3.0e+01 | 0.0e+00, 4.0e+01 | 7.1e-02, 6.4e+00 |
| Ochrophyta_Bacillariophyta | 0.0e+00, 4.7e+01 | 6.1e-01, 3.0e+01 | 4.1e+00, 9.8e+01 | 2.4e-01, 9.9e+01 | 0.0e+00, 3.2e+00 |
| Radiolaria | 0.0e+00, 4.3e+01 | 0.0e+00, 1.6e+01 | 0.0e+00, 1.8e+01 | 0.0e+00, 7.8e+01 | 8.2e-02, 2.1e+01 |
| Chlorophyta | 0.0e+00, 4.3e+01 | 5.3e-02, 7.9e+00 | 7.5e-03, 7.1e+00 | 0.0e+00, 4.5e+01 | 2.3e-03, 2.0e-01 |
| Cercozoa | 0.0e+00, 1.8e+00 | 2.5e-03, 5.6e+00 | 0.0e+00, 5.2e+00 | 0.0e+00, 2.8e+00 | 0.0e+00, 1.2e-01 |
| Haptophyta | 1.0e-02, 2.8e+01 | 1.2e-01, 4.7e+01 | 2.7e-02, 1.7e+00 | 0.0e+00, 3.0e+01 | 0.0e+00, 1.6e+00 |
| Sagenista | 5.0e-03, 1.3e+01 | 0.0e+00, 9.7e-01 | 0.0e+00, 2.6e-01 | 0.0e+00, 2.9e+00 | 1.2e-02, 2.9e+00 |
| Opalozoa | 0.0e+00, 3.3e+00 | 2.3e-02, 1.0e+00 | 0.0e+00, 4.9e-01 | 0.0e+00, 3.1e+00 | 1.0e-02, 5.0e+00 |
| Picozoa | 1.8e-01, 3.5e+01 | 3.8e-01, 1.1e+01 | 0.0e+00, 1.7e+00 | 0.0e+00, 4.6e+00 | 4.6e-01, 4.0e+01 |
| Ochrophyta_Chrysophyceae | 0.0e+00, 2.0e+00 | 0.0e+00, 9.7e-01 | 0.0e+00, 5.9e-01 | 0.0e+00, 1.7e+00 | 0.0e+00, 7.1e-01 |
| Telonemia | 0.0e+00, 8.1e-01 | 7.5e-03, 4.5e+00 | 0.0e+00, 2.2e-01 | 0.0e+00, 1.8e+00 | 2.8e-02, 6.6e-01 |
| Choanoflagellida | 0.0e+00, 7.4e+00 | 3.3e-02, 4.4e+00 | 0.0e+00, 1.7e+01 | 0.0e+00, 7.6e-01 | 1.4e-02, 5.1e+00 |
| Fungi | 0.0e+00, 2.8e-01 | 0.0e+00, 1.3e-01 | 0.0e+00, 6.5e-01 | 0.0e+00, 1.1e+00 | 0.0e+00, 4.0e-02 |
| Pseudofungi | 7.5e-03, 1.2e+01 | 1.7e-02, 7.4e+00 | 0.0e+00, 8.6e-01 | 0.0e+00, 3.7e+00 | 7.0e-03, 2.0e+01 |
| Ochrophyta_Bolidophyceae | 0.0e+00, 1.1e+00 | 0.0e+00, 5.3e-01 | 0.0e+00, 1.2e-01 | 0.0e+00, 7.4e-01 | 0.0e+00, 1.3e-01 |
| Cryptophyta | 0.0e+00, 5.5e+00 | 0.0e+00, 1.0e+01 | 0.0e+00, 4.2e-01 | 0.0e+00, 4.0e+00 | 0.0e+00, 1.1e+00 |
| Ochrophyta_Pelagophyceae | 0.0e+00, 9.1e+00 | 0.0e+00, 2.0e+00 | 0.0e+00, 1.2e+00 | 0.0e+00, 1.9e+00 | 0.0e+00, 1.6e+00 |
| Ochrophyta_Dictyochophyceae | 0.0e+00, 5.5e+00 | 0.0e+00, 2.2e+00 | 0.0e+00, 1.5e+00 | 0.0e+00, 9.2e-01 | 0.0e+00, 9.3e-02 |
| Stramenopiles_X | 0.0e+00, 6.0e-01 | 0.0e+00, 9.4e-01 | 0.0e+00, 2.5e-02 | 0.0e+00, 2.3e-01 | 0.0e+00, 8.6e-02 |
| Centroheliozoa | 0.0e+00, 2.1e+00 | 0.0e+00, 1.7e-01 | 0.0e+00, 4.8e+00 | 0.0e+00, 2.7e-01 | 0.0e+00, 2.1e-01 |
| Apicomplexa | 0.0e+00, 2.3e-01 | 0.0e+00, 1.3e+00 | 0.0e+00, 2.1e+00 | 0.0e+00, 2.5e+00 | 0.0e+00, 2.0e-02 |
| Katablepharidophyta | 0.0e+00, 1.4e+00 | 7.5e-03, 1.4e+01 | 0.0e+00, 2.7e-01 | 0.0e+00, 2.0e+00 | 0.0e+00, 1.8e+00 |
| Alveolata_X | 0.0e+00, 6.0e-01 | 0.0e+00, 7.5e-02 | 0.0e+00, 5.0e-03 | 0.0e+00, 1.3e-02 | 0.0e+00, 2.2e-02 |

| Division/Class | Size fraction ($\mu m$) | | | | |
|---|---|---|---|---|---|
| | 0.45-3 | 3-10 | 10-50 | 50-200 | 3-180 |
| Ochrophyta_MOCH-1 | 0.0e+00, 1.7e-01 | 0.0e+00, 7.0e-02 | 0.0e+00, 0.0e+00 | 0.0e+00, 2.5e-02 | 0.0e+00, 5.0e-02 |
| Ochrophyta_MOCH-2 | 0.0e+00, 9.3e-01 | 0.0e+00, 8.3e-01 | 0.0e+00, 1.0e-02 | 0.0e+00, 5.3e-01 | 0.0e+00, 1.6e-01 |
| Mesomycetozoa | 0.0e+00, 4.8e-02 | 0.0e+00, 4.8e-02 | 0.0e+00, 9.8e-02 | 0.0e+00, 1.2e-01 | 0.0e+00, 0.0e+00 |
| Dinoflagellata_Dinophyta_X | 0.0e+00, 5.8e-01 | 0.0e+00, 2.8e-01 | 0.0e+00, 3.0e-02 | 0.0e+00, 2.5e-02 | 0.0e+00, 1.4e-01 |
| Ochrophyta_Phaeophyceae | 0.0e+00, 1.6e-01 | 0.0e+00, 1.0e-01 | 0.0e+00, 3.3e-01 | 0.0e+00, 1.1e-01 | 0.0e+00, 0.0e+00 |
| Perkinsea | 0.0e+00, 2.5e-02 | 0.0e+00, 1.0e-02 | 0.0e+00, 0.0e+00 | 0.0e+00, 0.0e+00 | 0.0e+00, 0.0e+00 |
| Rhodophyta | 0.0e+00, 1.8e-01 | 0.0e+00, 1.8e-02 | 0.0e+00, 3.0e-01 | 0.0e+00, 2.1e-01 | 0.0e+00, 0.0e+00 |
| Dinoflagellata_Noctilucophyceae | 0.0e+00, 0.0e+00 | 0.0e+00, 0.0e+00 | 0.0e+00, 6.7e-01 | 0.0e+00, 0.0e+00 | 0.0e+00, 6.0e-02 |
| Opisthokonta_X | 0.0e+00, 1.5e-02 | 0.0e+00, 1.4e-02 | 0.0e+00, 7.5e-02 | 0.0e+00, 3.2e-01 | 0.0e+00, 5.4e-03 |
| Streptophyta | 0.0e+00, 7.5e-03 | 0.0e+00, 0.0e+00 | 0.0e+00, 5.0e-03 | 0.0e+00, 0.0e+00 | 0.0e+00, 8.0e-03 |
| Lobosa | 0.0e+00, 0.0e+00 | 0.0e+00, 0.0e+00 | 0.0e+00, 2.3e-02 | 0.0e+00, 0.0e+00 | 0.0e+00, 0.0e+00 |
| Apusomonadidae | 0.0e+00, 0.0e+00 | 0.0e+00, 0.0e+00 | 0.0e+00, 0.0e+00 | 0.0e+00, 0.0e+00 | 0.0e+00, 4.5e-03 |
| Conosa | 0.0e+00, 0.0e+00 | 0.0e+00, 0.0e+00 | 0.0e+00, 3.0e-02 | 0.0e+00, 0.0e+00 | 0.0e+00, 0.0e+00 |
| Dinoflagellata_Ellobiophyceae | 0.0e+00, 3.3e-02 | 0.0e+00, 2.2e-02 | 0.0e+00, 0.0e+00 | 0.0e+00, 0.0e+00 | 0.0e+00, 3.1e-02 |
| Discoba | 0.0e+00, 1.5e-02 | 0.0e+00, 0.0e+00 | 0.0e+00, 0.0e+00 | 0.0e+00, 0.0e+00 | 0.0e+00, 0.0e+00 |
| Metamonada | 0.0e+00, 0.0e+00 | 0.0e+00, 0.0e+00 | 0.0e+00, 1.0e-02 | 0.0e+00, 0.0e+00 | 0.0e+00, 0.0e+00 |
| Ochrophyta_MOCH-3 | 0.0e+00, 0.0e+00 | 0.0e+00, 1.0e-02 | 0.0e+00, 1.5e-02 | 0.0e+00, 0.0e+00 | 0.0e+00, 0.0e+00 |
| Ochrophyta_MOCH-4 | 0.0e+00, 6.5e-02 | 0.0e+00, 0.0e+00 | 0.0e+00, 0.0e+00 | 0.0e+00, 0.0e+00 | 0.0e+00, 0.0e+00 |