# Peer review of "An 18S V4 rDNA metabarcoding dataset of protist diversity in the Atlantic inflow to the Arctic Ocean, through the year and down to 1000 m depth"

_Earth System Science Data, 2021_

## Author Response (AR1)

**General comments from reviewer 1:**

General comments

In this paper, the authors constructed a metabarcoding dataset of marine protists communities in the Northern Svalbard region of the Arctic Ocean. These data comprise samples collected at some stations from the surface to the 1000 m depth every two to three months. The total number of amplicon sequence libraries is huge and the data would be enough to address a variety of ecological questions. Environmental metadata was also prepared for each of the sampling event. I agree that the dataset can be valuable as the study area is key to understand the connectivity between Arctic and Atlantic Oceans. However, I have some serious concerns regarding the sampling and sequencing strategies used in this study.

Major issues

In my opinion, sampling strategy is inappropriate to evaluate seasonal change or size- dependence of the microbial assemblages. For example, as the sampling locations are distributed among season, it would be difficult to decipher if the observed variation of microbe is owing to the season or just to the location. Similarly, size fractionation was not consistent across seasons. Samples were taken from 3-180 μm fraction in Jan to Mar, while 3-10, 10-50, and 50-200 μm fractions were applied for May to Nov. Sequencing platform was also inconsistent across samples (ie., MiSeq and HiSeq) and the different criteria were used for the downstream sequencing processing. Thus, the users will have difficulty in interpreting their results as they have to consider the possible effects of the different location, size fraction, and sequence platform. These methodological discontinuities would collectively diminish the overall quality of the dataset, and consequently the strength of the conclusion of the analysis.

The accuracy of the eukaryotic community profiling strongly depends on the choice of primer pair. I checked the ability of the primer set used in this study by in silico PCR analysis using a primer test tool (Silva TestPrime: https://www.arb-silva.de/search/testprime/), and found that the primer may amplify only 2% of known haptophyte sequences and 0.5% of Rhizaria sequences in the Silva SSU database. As these lineages are important components of marine microbial eukaryotes, the sequencing libraries were potentially biased due to the mismatches of some specific species. Although it is impossible to amplify all the rRNA genes in the environments, some primers have shown to be highly universal for both eukaryotes and prokaryotes (Parada et al.,
2015, 10.1111/1462-2920.13023; McNichol et al., 2021, 10.1128/mSystems.00565-21). These primer may be a standard method for monitoring overall communities of prokaryotes and eukaryotes across space and time. Although there is no golden standard in a choice of the primer pair so far, it's worth pointing out.

L8-9: Please specify size range of each of picoplankton, nanoplankton, etc.

L59-65 and the related metadata: The sampling conditions (location, depth and filter size) were bit complicated and sometimes inconsistent. For example, readers cannot recognize which size fractions and depths were applied for each season and site from this information (metadata is not suitable to see this kind of information). I recommend the authors to make a table summarizing the sampling site, season, depth and size fraction.

L88-89: A bit difficult to find out what the authors mean (i.e., "opposite patterns" of what?)

L129: The authors should justify the use of this primer set. As described earlier, this primer may not be a universal for some of the eukaryotic lineages, such as Haptophyta and Rhizaria.

L146-147: Please specify sequence length (e.g., 150 bp, PE) of MiSeq and HiSeq sequencing.

L203: It is difficult to know if there is a seasonality of eukaryotic communities from the Figure 4. I would recommend the authors to add an ordination plot such as NMDS to overview the variation of the community composition across season, size, and depths.

Environmental data file: Some parameters lack the unit (e.g., counting data of bacteria and virus).

Comments from reviewer 2:

The Arctic is undergoing major changes and data collection there is always challenging. Thus, Arctic datasets are of great value and interest. The manuscript of E. Egge and coauthors includes, as the title states, an 18S V4 rDNA metabarcoding dataset of protist diversity in the Atlantic inflow to the Arctic Ocean, through the year and down to 1000 m depth. This dataset is important and relevant. Yet, this dataset has major problems: there is inconsistency in the sampling points and the filtration procedures, as those change every month. There are also some other minor concerns in this version of the manuscript. Please see comments below.

-Concerns about the project: Samples were taken in 2014. Why is it important to publish the data, now in 2021, and in this journal? Is the dataset complete, or do you have more data from more recent samplings? May this dataset overlap with other datasets retrieved from other polar expeditions in this area?

-Concerns about the sampling stations: The distribution of the sampled stations seems chaotic . Why is not more consistent? Please explain the criteria for selecting the stations. Please provide information on the ice cover of each month (e.g. ice cover maps). What was the distance between the station and the ice cover? Please include this information in the dataset.

-Concerns about the size fractions: Why does the size of the fractions vary among months? This must be justified. Maybe the authors aimed to increase the resolution

of the plankton fractions in May, August and November. The only consistent and comparable fraction is the 0.45-3 μm fraction. This is a strong point of the dataset and should be highlighted in the article. A proposal to increase the robustness of the results would be to join the 3-10 μm, 10-50 μm and 50-200 μm samples into one (3-200 μm). In total, there would be only two size fractions (0.45-3 μm and 3-180/200 μm), but will be possible to compare the larger size fractions.

-Concerns about the sequencing: There is little consistency in the sequencing process. Why were the samples sent to two different sequencing centers? Why two different sequencing machines (HiSeq vs MiSeq)? Why were some samples replicated? Why was the number of PCR cycles changed in some samples? Etc.

-Concerns about the dataset: When the project is searched at the ENA browser (https://www.ebi.ac.uk/ena/browser/view/PRJEB40133) the result is: "No records were found for PRJEB40133." This means the project does not exist, and thus the dataset is not available. Dataset is 155 samples (140+15), and 199 sequencing events. At the ASV document, there are only 198 columns, so one sequencing event is missing. The environmental table includes many variables, but accessory information is lacking on what each variable means, and its units. The latitude and longitude is wrong at the first rows.

Specific comments:

-How to refer to samples: In this manuscript it happens that individual DNA samples (where each sample corresponds to a particular station, depth and fraction) are named as "sample_sizefract". This nomenclature is misleading because it appears to refer to a size fraction in general (and thus encompasses multiple samples). I propose to refer to all DNA samples simply as "sample" or as "DNA sample". Explain this in the text if necessary. To avoid further confusions, I recommend name any other type of sample (replica, niskin bottle, chlorophyll ...) with its particular specification.

-L.40: Please explain better about the challenges. Include if necessary, arguments about adverse weather.

-L.71-77: The information in this paragraph is confusing. If light is important, the dataset should include: sampling time, daylight hours, and number of hours of sunshine during the sampling day.

-Section 2.2.3: Please say that the results of Inorganic nutrients and Chlorophyll a are in Figure 2. Where is the methodology on chlorophyll measurements, cell counts, nutrients and other environmental variables?

-Section 3.1.1: Did you use a rosette? How many bottles per rosette? What time were the casts released?

-L.105-108: There is repeated information in this section. -

L.106, 107, 111: Why is the buffer temperature important? -

L.109: was the nylon mesh previously sterilized?

-L.124: "Subsequently 4 μL RNase was added". Why?

-L. 129-130: Please briefly comment on the advantage of this primer over others that are also commonly used.

-L.143-151: This section needs many clarifications, e.g:

-L.143-145: Why were the samples sent to two different sequencing centers? Why two different sequencing machines (HiSeq vs MiSeq)?

-L.146-147: What kind of issues did you have with Illumina Miseq in 2015, and why are they relevant? Why was it only done with two runs? Include "center" next to "GATC", since as it is it leads to confusion.

-L.148: What were these "few samples"? and why were replicas made?

-L.149-150: And why increasing the cycles was a solution? Why not pooling samples?

-L.151: Table 1 should include: type of platform used (Hiseq / MiSeq), site (Oslo / Germany), PCR cycles (25-30)...

-L.161-162: Are there versions of the PR2 databases? which one have you used? - L.168: better explain this "merged" and why.

-L.169: Why doesn't the 3-180 µm fraction have 40,000 reads, as the others? Why 3-180 µm fraction in Figure 3 has variable number of reads?

-L.170: please indicate if this subsampling was made with a specific function in R

-L.176-178: For consistency, previous sections should explain where the sequencing data is (ENA, link...).

-L.181: I guess "44 samples from niskin bottles" is more accurate (instead of "44 Niskin samples")

-L.182: at this point the reader cannot understand what this code means: "May_P4_net_10_50 failed"

-L.182: "These samples are in the following referred to as 'sample_sizefract".Where?

-L.181, 185, 187 and others: change "sample_sizefract" to "sample" or "DNA sample". See general comments.

-L.185: please explain what is ENA.

-L.187: why they were merged? Please explain this.

-L.191: the removal of singletons was not mentioned in section 4.2

-L.192 and 193: I imagine that when you say "size fraction" you mean "sample".

-L.194: I recommend to separate the numbers with ";" instead of "," .

-L.195-196: Please clarify this point.

-L.200: how was the slope calculated?

-L.202: why is this correlation important?

-L.204-205: both figures, Figure 4 and Figure A1, show: metabarcoding reads, ASV, division and class levels. Please rephrase.

-L.206-207: Is the name of the fractions (pico-, nano- and micro-) important? Then explain them at the beginning of the manuscript, and include them in datasets and graphs.

-L.209: Please explain why in Table A2 some groups do not have any >0 values.

-L.210: Please explain which groups are heterotrophic or parasitic.

-L.211: At this point data is "relative abundances", not "read abundances".

-L.210-255: please explain where the values of "richness" comes from.

-L.219: please explain which groups are phototrophs

-L.227: please explain which Division / class corresponds to Diatoms

-L.257: This dataset is descriptive and does not include patterns or dynamics.

-L.258: This dataset is not about food webs.

-L.256-260: An interesting argument to add is that this dataset is a baseline for future studies aiming to determine temporal changes.

-L.262: https://www.ebi.ac.uk/ena/browser/view/PRJEB40133 says: No records were found for PRJEB40133.

-L.263: I recommend to remove this part "corresponding to the size-fractionated plankton samples"

-All figures and tables: captions need improvement. Should provide more information and guidance. Here some comments:

-Figure 2: some profiles are missing, e.g.: there are at least 6 stations in March, and at least 4 go deep. Something is wrong with the y-axis: Is it logarithmic?

-Figure 3: change "sample_sizefract" by "sample" or "DNA sample". Figure caption needs improvement. The reader here is lost. Avoid using references like "asvtab3_merged_subsamp_readnum.txt". Better if you make other types of references, for example: "see details at Table X ..."

-Figure 4: remove codes and put an understandable legend (e.g. "1m" instead of 0001). If in the bars, the order of protist groups is from left to right, the same order should be in the color code of the legend, but it is the other way around. The two fractions 3-180 µm and 3-10 µm are very different, and should appear separately (different columns).

-Figure A1: this figure needs to be linked with the main text.

-Table A1: include the N (number of samples) included in each size fraction.

-Table A2: Please explain why in Table A2 some groups do not have any >0 values.

####################

**Response to reveiwer 1:**
Dear reviewer. Thanks for taking the time to thoroughly review our paper.

In Arctic waters, processes connected with the ice edge such as melting and stratification, are crucial for the primary production and microbial life. Therefore, we aimed to get as close as possible to the ice edge during each cruise, rather than sampling from a fixed location. The temporal change in taxonomic composition in our data reflects what is previously known about the succession of the major groups of microplankton in the Arctic – few autotrophs in winter, a spring bloom dominated by diatoms, *Phaeocystis* and *Micromonas*, followed by a high proportion of dinoflagellates in summer. All sampling locations are beyond the polar circles and therefore experience polar night, i.e. the sun is below the horizon during some period from autumn until spring. This successional pattern could thus not have been produced only by variation in location.

We agree that the inconsistent size-fractionation is unfortunate. This was due to the logistical constraints and different space available to us on the vessels used for these cruises. However, we consider it unlikely that the taxonomic composition in the 3-180 µm fraction, if it were also sampled in May, August and November would be qualitatively different from the composition in the 3-200 µm fraction (consisting of the 3-10, 10-50 and 50-200 fraction merged). We think that it is better to present the original data than to create an artificial 3-200 µm fraction, as it is an open question how such a fraction should be calculated (e. g. by taking the average of the fractions, or weighting each fraction by the size of the fraction, due to the larger genomes and higher rDNA copy numbers of larger cells (see e.g. Prokopowich et al. 2003 "The correlation between rDNA copy number and genome size in eukaryotes" Genome, 50, 48-50.) If the reader wants to see what such a fraction would look like they are free to transform the data as they please.

Regarding different sequencing platforms and downstream processing – in the cases where the same sample was sequenced with both methods, the taxonomic composition was similar between the two platforms. This can be seen in the interactive version of Figure 4 found in the Shiny app, where the taxonomic compositions produced by MiSeq and HiSeq are shown separately for the corresponding samples. We are therefore confident that variation in sequencing method does not introduce a bias. The reason for using two different sequencing methods are now explained in the manuscript (L 177-185).

With regards to primer choice, it is always a question whether to use published primers that are already tested, or to try to improve on previous primers, but risking that the changes make them less efficient. The primers from Piredda et al. (2017) are an improvement over the very widely used primer developed by Stoeck et al (2010) which have been used in more than 60 studies. The Piredda primers aim of reducing the biases against Haptophyta seen in the Stoeck primers. They have been carefully tested. Allowing for 1 mismatch, they amplify 94% of Haptophyta, and 58% of Rhizaria. They have been used in at least 10 studies including for the Ocean Sampling Day project (Tragin, M., Zingone, A., & Vaulot, D. 2018. Comparison of coastal phytoplankton composition estimated from the V4 and V9 regions of the 18S rRNA gene with a focus on photosynthetic groups and especially Chlorophyta. Environmental Microbiology, 20, 506 520. https://doi.org/10.1111/1462-2920.13952). However, we agree that possible primer bias should be kept in mind when interpreting the data.

Please keep in mind that the purpose of this paper is purely to present data. We are not concluding anything regarding the seasonal and depth variation in taxonomic composition. These types of analyses will be presented in follow-up papers. We have now edited the abstract to be more descriptive and less conclusive.

Specific comments

L8-9: Please specify size range of each of picoplankton, nanoplankton, etc.
A: This is now done.

L59-65 and the related metadata: The sampling conditions (location, depth and filter size) were bit complicated and sometimes inconsistent. For example, readers cannot recognize which size fractions and depths were applied for each season and site from this information (metadata is not suitable to see this kind of information). I recommend the authors to make a table summarizing the sampling site, season, depth and size fraction.
A: Thank you for this suggestion. Such a table is now provided (Table 2)

L88-89: A bit difficult to find out what the authors mean (i.e., "opposite patterns" of what?)
A: It is now specified that we mean "inversely related to each other", L103

L129: The authors should justify the use of this primer set. As described earlier, this primer may not be a universal for some of the eukaryotic lineages, such as Haptophyta and Rhizaria.
A: See response regarding primer pair above.

L146-147: Please specify sequence length (e.g., 150 bp, PE) of MiSeq and HiSeq sequencing.
A: Done.

L203: It is difficult to know if there is a seasonality of eukaryotic communities from the Figure 4. I would recommend the authors to add an ordination plot such as NMDS to overview the variation of the community composition across season, size, and depths.
A: We consider NMDS or other ordination analyses as outside the scope of a data paper, as these types of analyses are done to infer patterns in the data.

Environmental data file: Some parameters lack the unit (e.g., counting data of bacteria and virus).
A: The environmental data file is now edited. Units of the parameters are now specified in Table 1 in the manuscript.

Dear reviewer (2). Thanks for taking the time to thoroughly review our paper.

Response to your main comments:

Time gap between sampling and publishing: We understand that it seems strange to wait seven years to publish. However, the SEANOE dataset was dated 2014 because that is the year the samples were taken, but the data were processed later. Clearly data from 2014 are still valuable in 2021, especially in the light of the rapid changes occurring in Arctic marine ecosystems, and their publication is justified. This journal was chosen because the format suits our study, and we thought the data are relevant for this special issue. Regarding overlap with other studies from the Arctic, there are certainly other metabarcoding datasets targeting protists from the Arctic waters surrounding Svalbard, but to our knowledge, our study is unique in that it comprises all the seasons, and depths down to 1000 m. Moreover, our data will serve as a reference dataset for more recent work and may help detect changes due to processes such as the "atlantification". Please note that such a delay in publication is not unusual with cruises in polar regions. For example the MALINA cruise was performed in 2009 and the paper presenting the data was published in this journal in 2021 (Massicotte et al. 2021. The MALINA oceanographic expedition : How do changes in ice cover, permafrost and UV radiation impact biodiversity and biogeochemical fluxes in the Arctic Ocean? Earth System Science Data, 13, 1561  1592. https://doi.org/10.5194/essd-13-1561-2021).

Choice of sampling stations: Sampling in the Arctic is constrained by weather and ice cover, among other things. Sampling needs to take place when ice and weather permits. As pointed out in the answer to reviewer 1, it was important to get as close as possible to the ice edge, due to the important primary production and microbial processes taking place there, especially in the spring and summer cruises. Map of ice cover can be found in Wilson et al. 2017, referenced in the paper. This type of sampling is also extremely expensive, therefore the MicroPolar paired with the project CarbonBridge (e.g. Randelhoff et al. 2018, referenced in the paper), which further constrained the choice of sampling sites.

Size fractions: As explained in the response to reviewer 1, the different size fractionation regimes between January and March, and the other months was due to the different space available to us on the vessels used on these cruises. However, we consider it unlikely that the taxonomic composition in the 3-180 micron fraction, if it were sampled in the other months, would be qualitatively different from the composition in the 3-200 fraction (consisting of 3-10, 10-50 and 50-200).

Sequencing: There was a world-wide problem with the delivery of the MiSeq reagents in the spring of 2015. Therefore, we initially had to rely on a modified HiSeq protocol (according to the GATC sequencing centre, this was their replacement for

the MiSeq protocol as long as the problems persisted.) By the time we had received our HiSeq reads, and we wished to increase the sequencing coverage for certain samples, the MiSeq problems were resolved, and we sequenced the samples at a local sequencing centre. In the cases where the same sample was sequenced with both methods, the taxonomic composition was similar between the two platforms. This can be seen in the interactive version of Figure 4 found in the Shiny app, where the taxonomic compositions produced by MiSeq and HiSeq are shown separately for the corresponding samples (`https://micropolar-protists.metapr2.org/`). We are therefore confident that variation in sequencing method does not introduce a bias.

Regarding the ENA/SRA accession: thank you for your attention to this, the raw read files deposited in ENA/SRA have now been released.

Regarding the discrepancy between the number of rows in the metadata file and the number of columns in the ASV-table, you are right. The sample in question originally had a low number of reads, and no reads remained after DADA2 processing, therefore it was automatically removed from the processed ASV-table. We still chose to keep the metadata information for the raw read file, as it will be available from the ENA. We have added a note about this in the caption of Table 4.

Regarding the environmental data, these data have now been edited to match the table describing these data in the manuscript. We double-checked the latitude and longitude, and could not find any errors.

**Response to specific comments, reviewer 2**

L.40: Please explain better about the challenges. Include if necessary, arguments about adverse weather.
A: This has now been specified on L41-42: "Arctic winter microbial eukaryote communities are particularly understudied due to logistic challenges which include ice cover and frequent storms.."

-L.71-77: The information in this paragraph is confusing. If light is important, the dataset should include: sampling time, daylight hours, and number of hours of sunshine during the sampling day.
A: The dataset now includes daylight hours, and euphotic zone depth of the spring and summer cruises. (section 2.2.1) Sampling time and number of hours of sunshine are unfortunately not available.

-Section 2.2.3: Please say that the results of Inorganic nutrients and Chlorophyll a are in Figure 2. Where is the methodology on chlorophyll measurements, cell counts, nutrients and other environmental variables?

A: Reference to Figure 2 has been added to this section. The methodology on chl a, nutrients and cell counts has been described briefly, citing the papers where the detailed methodology can be found. See sections 2.2.4 and 2.2.5.

-Section 3.1.1: Did you use a rosette? How many bottles per rosette? What time were the casts released?

A: Yes, we used a 12-bottle rosette, this is now specified on L125. The release time of the casts is unfortunately not available.

-L.105-108: There is repeated information in this section. -

A: No, the second time the filter is mentioned is after the material is washed off the filter. The filter itself was also preserved in buffer, to isolate DNA from any material stuck on or in the filter.

L.106, 107, 111: Why is the buffer temperature important?

A: Heating the buffer to 65 degrees facilitates release of DNA from the material into the buffer. Incubation at 65 degrees is also recommended in the Qiagen protocol, therefore we did not find it necessary to explain in the text.

L.109: was the nylon mesh previously sterilized?

A: It was rinsed with dH2O between each sample, which is now stated in L129

-L.124: "Subsequently 4 μL RNase was added". Why?

A: To ensure the isolated DNA is pure and free of RNA, which may interfere in downstream proceses. This is a standard step of most DNA isolation protocols, therefore we did not find it necessary to explain in the text.

-L. 129-130: Please briefly comment on the advantage of this primer over others that are also commonly used.

A: This is now done on L160-162

-L.143-151: This section needs many clarifications, e.g:

-L.143-145: Why were the samples sent to two different sequencing centers? Why two different sequencing machines (HiSeq vs MiSeq)?

-L.146-147: What kind of issues did you have with Illumina Miseq in 2015, and why are they relevant? Why was it only done with two runs? Include "center" next to "GATC", since as it is it leads to confusion.

A: Please see answer to referee #1

-L.148: What were these "few samples"? and why were replicas made?

A: This was done to assess variation between DNA extracts and annealing temperature (explained in L180-183). The samples in question are indicated in the meta data file.

-L.149-150: And why increasing the cycles was a solution? Why not pooling samples?

A: To obtain higher concentration of the PCR product. We realised that even with pooling+concentrating we would still likely get too little PCR product.

-L.151: Table 1 should include: type of platform used (Hiseq / MiSeq), site (Oslo / Germany), PCR cycles (25-30)...
A: Sequencing platform is already included. All HiSeq was done in Germany, all MiSeq in Oslo, so that information is redundant. Information on the number of PCR cycles is now included.
-L.161-162: Are there versions of the PR2 databases? which one have you used?
A: PR2 database version is now specified L198
-L.168: better explain this "merged" and why.
A: Two files that contain sequences from the same DNA sample were merged by taking the sum of the read number for each ASV. The merging was done to increase the read number in the samples that initially had a low read number after HiSeq sequencing. (L205-206)

-L.169: Why doesn't the 3-180 µm fraction have 40,000 reads, as the others? Why 3-180 µm fraction in Figure 3 has variable number of reads?
A: Because it comprises a wider range of cell sizes than the other fractions. The subsample was taken to be the sum of the subsamples from the 3-10, 10-50 and 50-200 µm fractions. However, in some 3-180 µm samples the read number was less than 88,000. (L208-209).

-L.170: please indicate if this subsampling was made with a specific function in R
A: Yes, this is now stated on L211.

-L.176-178: For consistency, previous sections should explain where the sequencing data is (ENA, link...).
A: To avoid repetition, these lines have instead been moved to the Code and data availability section.

-L.181: I guess "44 samples from niskin bottles" is more accurate (instead of "44 Niskin samples")
A: This has been changed.
-L.182: at this point the reader cannot understand what this code means: "May_P4_net_10_50 failed"
A: It is now specified which sample we are referring to.
-L.182: "These samples are in the following referred to as 'sample_sizefract".Where?
A: This has been changed.
-L.181, 185, 187 and others: change "sample_sizefract" to "sample" or "DNA sample". See general comments.
A: This is now done.
-L.185: please explain what is ENA.
A: Done (L227-228)

-L.187: why they were merged? Please explain this.

A: Two files that contain sequences from the same DNA sample were merged by taking the sum of the read number for each ASV. The merging was done to increase the read number in the samples that initially had a low read number after HiSeq sequencing. This is now explained on line (205-207).

-L.191: the removal of singletons was not mentioned in section 4.2

A: The dada2 pipeline does not remove singleton per se but do not construct ASVs that are supported by a single read (see https://github.com/benjjneb/dada2/issues/320). We have now removed the word "singleton" since this is part of the standard dada2 processing.

-L.192 and 193: I imagine that when you say "size fraction" you mean "sample".

A: No, we mean what it says in the text.

-L.194: I recommend to separate the numbers with ";" instead of "," .

A: Done.

-L.195-196: Please clarify this point.

A: The fractions were not obtained from the same number of samples, since e.g. 3-180 was only taken in January and March, whereas 3-10 -- 50-200 were only taken in May, August and November. Explained on L238-239

-L.200: how was the slope calculated?

A: This is now explained in the Methods section, L215

-L.202: why is this correlation important?

A: Because if e.g. samples with very low ASV richness (i.e. number of ASVs) always had a high slope, i.e. the curve was very steep at the endpoint, this would likely mean that the number of reads obtained from these sample was not high enough, and not that they had low richness (L245-246)

-L.204-205: both figures, Figure 4 and Figure A1, show: metabarcoding reads, ASV, division and class levels. Please rephrase.

A: Unfortunately we are not sure what is meant here.

-L.206-207: Is the name of the fractions (pico-, nano- and micro-) important? Then explain them at the beginning of the manuscript, and include them in datasets and graphs.

A: The terms are now explained in the abstract and on L250-252.

-L.209: Please explain why in Table A2 some groups do not have any >0 values.

A: Some groups had very low proportional abundance. These numbers are now changed to scientific notation.

-L.210: Please explain which groups are heterotrophic or parasitic.

A: Now done on L254

-L.211: At this point data is "relative abundances", not "read abundances".

A: Correct, thanks for paying attention to this. This has now been corrected.

-L.210-255: please explain where the values of "richness" comes from.

A: Now explained on L212-213

-L.219: please explain which groups are phototrophs

A: Done, L262.

-L.227: please explain which Division / class corresponds to Diatoms

A: Now explained in the captions to Figure 4.

-L.257: This dataset is descriptive and does not include patterns or dynamics.

A: True, this sentence has been edited

-L.258: This dataset is not about food webs.

A: True, this sentence has been edited

-L.256-260: An interesting argument to add is that this dataset is a baseline for future studies aiming to determine temporal changes.

A: Good point, this is now included.

-L.262: https://www.ebi.ac.uk/ena/browser/view/PRJEB40133 says: No records were found for PRJEB40133.

A: This project has now been released by ENA.

-L.263: I recommend to remove this part "corresponding to the size-fractionated plankton samples".

A: This is now done.

-All figures and tables: captions need improvement. Should provide more information and guidance. Here some comments:

-Figure 2: some profiles are missing, e.g.: there are at least 6 stations in March, and at least 4 go deep. Something is wrong with the y-axis: Is it logarithmic?

A: From some stations data from only a limited part of the water column was available. This is now pointed out in the caption.

We discovered errors in the map showing the sampling stations, i. e. it showed two stations that were not sampled for protist metabarcoding; M01 and N01. This has been corrected. The number of stations in Figure 1 and 2 are now the same.

-Figure 3: change "sample_sizefract" by "sample" or "DNA sample". Figure caption needs improvement. The reader here is lost. Avoid using references like "asvtab3_merged_subsamp_readnum.txt". Better if you make other types of references, for example: "see details at Table X

A: Done.

-Figure 4: remove codes and put an understandable legend (e.g. "1m" instead of 0001). If in the bars, the order of protist groups is from left to right, the same order should be in the color code of the legend, but it is the other way around. The two fractions 3-180 µm and 3-10 µm are very different, and should appear separately (different columns).

A: Figure 4 has been edited accordingly.

-Figure A1: this figure needs to be linked with the main text.

A: Done (L249)

-Table A1: include the N (number of samples) included in each size fraction.

A: Done.

-Table A2: Please explain why in Table A2 some groups do not have any >0 values.

A: Some groups had very low proporition of reads, the numbers are now changed to scientific notation to avoid only 0 values for some groups.